# Influence of Storage Conditions on the Stability of Gum Arabic and Tragacanth

**DOI:** 10.3390/molecules27051510

**Published:** 2022-02-23

**Authors:** Paweł Ramos, Mateusz Broncel

**Affiliations:** 1Department of Biophysics, Faculty of Pharmaceutical Sciences in Sosnowiec, Medical University of Silesia in Katowice, Jedności 8, 41-200 Sosnowiec, Poland; mateusz_broncel@interia.pl; 2Doctoral School, Medical University of Silesia in Katowice, Jedności 8, 41-200 Sosnowiec, Poland

**Keywords:** gum arabic, tragacanth, proper storage conditions, UV-irradiation, higher temperature, relative humidity, TGA, c-DTA, colorimetric analysis, UV spectrophotometry, optical microscope observations

## Abstract

Storage conditions should be chosen so that they do not affect the action and stability of the active pharmaceutical substance (API), and excipients used in pharmacy. UV irradiation, increased temperature, and relative humidity can decompose storage substances by photolysis, thermolysis, and hydrolysis process, respectively. The effect of physical factors may be the decomposition of pharmaceutical substances or their inappropriate action, including pharmacological effects. Polymers of natural origin are increasingly used in the pharmaceutical industry. With this in mind, we evaluated the effect of storage conditions on the stability of gum arabic (GA) and tragacanth (GT). The influence of higher temperature, UV irradiation, and relative humidity on GA and GT was tested. Thermogravimetry (TG, c-DTA), colorimetric analysis, UV-Vis spectrophotometry, and optical microscopy were used as research methods. The TGA and c-DTA examination indicated that decomposition of GA starts at a higher temperature compared to GT. This indicate that gum arabic is more resistant to higher temperatures compared to tragacanth. However, the conducted analysis showed that gum arabic is more sensitive to the tested storage conditions. Among the tested physical conditions, both polymers were most sensitive to conditions of increased relative humidity in the environment.

## 1. Introduction

Proper storage conditions are a very important aspect that affects the action and stability of the active pharmaceutical substance (API), and excipients used in pharmacy. Physical factors such as temperature, UV radiation, or moisture may adversely affect the stored pharmaceutical preparation [1,2,3,4,5,6,7,8,9]. As a result of the higher temperature, chemical bonds may be broken in the thermolysis process, which leads to the degradation of the stored substance [1,2,3]. As a result of exposure to UV radiation, energy quanta may be absorbed by the chemical substance, which may lead to its destruction in the process of photolysis [4,5,6]. In addition, improperly selected air moisture during the storage of API or excipients may also adversely affect the stability of the chemical substance [7,8,9]. Substances characterized by high hygroscopicity are particularly sensitive to increased moisture [8,9]. The active pharmaceutical ingredient or excipients may be decomposed by the hydrolysis process by the action of steam or water [10]. The derivatives of carboxylic acids (esters, lactones, lactams, amides) show the greatest sensitivity to this type of reaction [11]. Moisture can also influence on the crystalline system of APIs [12,13]. This applies especially to unstable or metastable amorphous forms which, under the influence of increased humidity, tend to change the internal order consisting of reaching the form with the lowest internal energy [12]. This may alter the release profile of the drug or influence on bioavailability of the drug substance [11,13]. Increased moisture sorption by the finished drug formulation may cause an increase in microbial contamination [14].

Substances also sensitive to increased humidity or light can be decomposed by oxidation [15]. Particularly sensitive to oxidation reactions are thiols, thioethers, ethers, and aldehydes, as well as natural gums [11]. Oxidation processes are often accompanied by a change in color or odor, which makes them easier to observe [11].

Polymers of natural origin are increasingly used in the pharmaceutical industry. This is due to their many advantages, such as low production cost, high occurrence in nature, and good biodegradability, which protects the environment [16,17]. Natural polymers can be used in pharmacy in polymeric drug delivery systems [18], as thickening agents [19], drug release rate controlling agents [19], taste-masking agents [20], and protective or stabilizing agents [21]. Examples of natural polymers that are used in pharmacy are gum arabic (GA) and tragacanth (GT) [11,22].

Gum arabic (GA) is a substance obtained naturally by cutting *Acacia Senegal* shoots [11,20,22,23]. Spray-dried acacia is used as a pharmaceutical raw material [11]. GA is a complex of polysaccharides and hemicelluloses with a molecular weight of 240–580 kDa. Gum arabic is well soluble in glycerol and propylene glycol. It is practically insoluble in ethanol [11,22]. It dissolves very slowly in water in a ratio of 1 to 2.7. Arabine is the main ingredient of gum arabic [24]. In pharmacy, GA is used in oral and topical drug forms mainly as an emulsifying and viscosity increasing agent [20]. Arabic gum has good bioadhesive properties. It can be used in the production of soft tablets and tablets as a binder [11,22].

Gum tragacanth (tragacanth) (GT) is a substance obtained naturally by cutting the shoots of *Astragalus gummifer* or another species of the genus *Astragalus* [11,20,22]. Tragacanth is an odorless and tasteless white or yellow powder. GT is practically insoluble in water, ethanol, and other inorganic solvents [11,22]. It contains a mixture of soluble and insoluble polysaccharides, mainly basorin and tragacanthines. Tragacanth gels in water and is used as an emulsifier and dispersant in creams, emulsions, and gels [20,25]. In pharmacy, it is also used as a binder in the granulation process [11].

The aim of this study was to evaluate the effect of storage conditions on the stability of gum arabic and tragacanth. The influence of higher temperature (40, 50, 60, 70, and 80 °C), the influence of UV irradiation (30, 60, 90, and 120 min), and relative humidity (RH 45%, 65%, and 75%) on the tested polymers of natural origin were investigated. Thermogravimetry (TG, c-DTA), colorimetric analysis, UV-Vis spectrophotometry, and optical microscope imagine were used as research methods.

## 2. Results and Discussion

### 2.1. TG and c-DTA Dynamic Analysis

Thermogravimetric curves of gum arabic (GA) presented in Figure 1 showed that the thermally decomposition is stated at 284.1 °C and contain two stages. The mass loss in first and second stages were −6.30% and −54.97%. The DTG curve of GA presented two peaks corresponding with TG curve (Figure 1). DTG first stage of mass loss occurred in temperature ranged from 41.5 to 138.7 °C with the maximum peak in 79.7 °C. This stage is related with water release and it is in good agreement with the literature [26]. The water release was accompanied by an endothermic peak on the c-DTA curve with maximum in 87.9 °C (Figure 1). The last stage began at a temperature of 229.8 °C with maximum peak mass loss in 315.8 (−9.99%/min.) and was associated with a degradation of gum arabic [26,27,28]. The second stage was accompanied by an exothermic peak recorded on the c-DTA curve with the maximum in 318.5 °C (Figure 1).

The TG curves of gum tragacanth (GT) presented in Figure 2 showed that the thermally decomposition begins at 238.2 °C and contains two stages. The mass loss in the first stage was −9.39%, and second stage was −63.30%, respectively. The DTG curves of GT presented two peaks corresponding with TG curves (Figure 2). The DTG first stage of mass loss occurred in the temperature range between 31 and 126.4 °C with the maximum peak in 67.3 °C. This stage is related with water release [28]. The water release was accompanied by an endothermic peak on the c-DTA curve with a maximum of 58.9 °C (Figure 2). The second stage occurred in the temperature range of 211.3 to 389.4 °C with maximum peak in 258.9 °C and mass change −8.29%/min. The second stage is related to the decomposition of tragacanth [28]. The second stage was accompanied by an exothermic peak recorded on the c-DTA curve with a maximum of 258.5 °C (Figure 2).

The performed analysis showed that gum arabic is more resistant to higher temperatures. The onset decomposition for gum arabic is a delay of 45.9 °C compared to tragacanth.

### 2.2. TG Isothermal Analysis

For initial samples of gum arabic (Figure 3a) and tragacanth (Figure 3b), isothermal thermogravimetric measurements were done. The measurements were prepared in the temperature range 40–80 °C. The analysis lasted 60 min for each temperature. The obtained maximum mass loss for GA and GT was different and depended on temperature. The greatest mass losses in isothermal conditions for gum tragacanth and gum arabic were recorded at temperatures of 50 and 60 °C, respectively (Figure 3).

The mass loss in the studied temperature range is related with water release [26,28]. The greater mass loss occurring at a lower temperature for GT is associated with a greater initial amount of adsorbed water and an earlier first stage of decomposition compared to GA (Figure 2) [26,27,28].

Gum arabic and tragacanth were stored in different relative humidity (RH) conditions. In the experiment 45%, 65%, and 75% RH were used. GA and GT stored in a different combination of condition (RH/temperature/time) were tested. Gum arabic (Figure 4a) and tragacanth (Figure 4b) were exposed to different RH conditions isothermal TG measurements. For each sample, measurements at 50 °C for 60 min were performed.

Analyzing the initial samples of GA and GT, it was observed that the samples were more than twice as large as the mass loss for tragacanth. This fact also corresponds with the recorded GT curves (Figure 1 and Figure 2) and indicated that the tragacanth content absorbed water more. This was also observed in other works [28,29].

For gum arabic, reduction in the mass loss was observed after 3 days of storage in 45% RH at 25 °C. This was due to the initial loss of absorbed water by the GA. However, after 2 months of storage in 45% RH and 25 °C, the amount of absorbed water almost doubled compared to initial GA (Figure 4a). For GA storage in 65% RH at 40 °C, gradual mass loss with an increase in day storage was observed. This was related to higher temperature storage. Similar results were observed by Zografi et al. in the poly(vinylpyrrolidone) studies [30]. They recorded less water sorption for the polymer stored at a higher temperature [30]. This fact corresponded well with other measurements [31]. For GA after 3 days of storage in 75% RH at 25 °C, a very large mass loss was observed. This is due to the high water absorption under these conditions [31]. For gum arabic, storage at a higher temperature (40 °C) with increased relative humidity, compared to room temperature (25 °C) with normal and increased relative humidity, caused a decrease in the amount of water in the sample. Similar results for Acacia gum were reported in another work [32]. A lower content of water with an increasing storage temperature was recorded for Acacia gum [32]. Mansoori et al. explained this phenomenon, suggesting that with higher temperatures, intermolecular forces of attraction diminish because of the increased kinetic energy of molecules. This results in increased molecular mobility. For that reason, water molecules, which have nonlimited motions, are expected to form weaker bonds with the binding sites of polymer molecules [32].

For gum tragacanth, gradual mass loss with increased days of storage in 45% RH at 25 °C was observed compared to gum arabic. In addition, for the storage condition of 65% RH and 40 °C a slight mass loss was observed. However, in GT storage with 75% RH at 255 °C, a very large mass loss was observed (Figure 4b). This fact was due to the high water absorption through tragacanth with a large amount of relative humidity in storage conditions [33].

### 2.3. Colorimetric Analysis

A colorimetric analysis in the CIE L*a*b* color space system was performed for the initial samples, stored under UV radiation, higher temperature, and under a combination of various conditions of relative humidity and temperature.

The colorimetric analysis for tested natural polymers were similarly changes after UV-irradiation. For gum arabic, the total color difference (ΔE*) parameter ranged from 0.58 to 1.47 (Table 1). These values indicate a barely or faintly visible color change. The L* parameter took on slightly lower values for UV-irradiated GA compared to the initial sample. This indicates a slight darkening of gum arabic under the influence of UV radiation. Kuan et al. noticed a similar dependence, explaining the phenomenon of oxidative degradation of gum arabic [34].

For gum tragacanth, the total color difference (ΔE*) parameter ranged from 0.14 to 2.32 (Table 2). These values indicate no or a faintly visible color change. For GT, this parameter did not change linearly with the increasing time of exposure to ultraviolet. In another work, poly(vinyl chloride) and PVC membranes containing cadmium pigments were exposed to UV irradiation [35]. Minimal changes in the color of tested synthetic polymer was observed [35].

The colorimetric analysis in CIE L*a*b* system for gum arabic and gum tragacanth showed similar changes after temperature treatment. For GA and GT, the total color difference (ΔE*) parameter ranged from 0.35–1.17 (Table 1) and 0.38–0.77 (Table 2) respectively. These values indicate no or a barely visible color change [36]. A minimal change in total color difference (ΔE*) parameter for polylactide (PLA) treated with 100 °C was recorded by other researchers [37]. The same researchers registered a large change in the ΔE* parameter for polyhydroxyalkanoate (PHA), which indicates the high sensitivity of this polymer to exposure on higher temperature [37].

Gum arabic and tragacanth samples stored under conditions of different relative humidity and temperature contents were also evaluated for color change in the CIE L*a*b* system. For these samples, the change in all color parameters assessed was the most apparent. The largest change was recorded for GA and GT stored under the conditions of 75% RH/25 °C/3 days and 65% RH/40 °C/6 days (Table 1 and Table 2). This change was caused by the increased relative humidity in the environment and the associated water sorption by the tested samples. Increased water sorption in these conditions by gum arabic and tragacantha was confirmed by isothermal thermogravimetric analysis (Figure 4.).

### 2.4. UV Spectrophotometry Analysis

In the work, all tested samples were also analyzed with the use of UV-Vis spectrophotometry. For this purpose, the absorbance spectra in the UV range were recorded (Figure 5).

UV spectra obtained in range 200–380 nm for gum arabic treated UV-irradiation, different temperature storage (40–80 °C) during 60 min, and relative humidity were similarly (Figure 5a). Relative to the initial sample of GA, a gently hypochromic effect manifested in a decrease in the intensity of the absorption band at maximum wavelength 273 nm was observed. In the vicinity of a wavelength of 273 nm, it shows an absorbance of tryptophan, which was proven in other studies [38]. Three fractions (F1–F3) from *Acacia senegal* gum were isolated in these studies. Fractions 1, 2, and 3 were arabinogalactan peptide, arabinogalactan protein, and glycoprotein respectively. Fractions 2 and 3 showed the same maximum absorbance peak as pure gum arabic [38]. Fractions 2 and 3 contained tryptophan [38]. Tryptophan is very sensitive to UV radiation and is degraded by it with the formation of hydroxyl radical (^•^OH) [39]. This fact was confirmed by EPR spectroscopy [39]. Additionally, the elevated temperature intensifies this effect [39,40]. Therefore, the decrease in absorbance in this range under the influence of UV and higher temperature can be explained by the decomposition of tryptophan. The hypochromic effect was most visible for gum arabic storage in 75% (RH)/25 °C/3 days. The greatest changes were also observed for this sample by other techniques (Figure 4, Table 1). The deterioration in the solubility of GA in water may be responsible for this effect.

UV spectra obtained in the range of 200–380 nm for gum tragacanth treated and tested in the storage conditions were similar (Figure 5b). Relative to the initial sample, only for GT in the sample storage at 75% (RH)/25 °C/3 days did a hyperchromic effect manifest an increase in the intensity of the absorption band at maximum wavelength 270 nm. This fact may be due to the improving the solubility of GT in water. The greatest changes were also observed for this sample by other techniques (Figure 4, Table 2).

### 2.5. Optical Microscopy Observations

For tested natural polymers, optical microscope images were recorded (Figure 6). Among the tested samples, the largest changes in the microscopic images for gum arabic, and tragacanth stored under the conditions of 75% (RH)/25 °C/3 days were recorded (Figure 6f). These changes for gum arabic were also visible macroscopically. Gum arabic stored under the conditions of 75% (RH)/25 °C/3 days changed its form from a loose, white powder to a yellow, compact form resembling a crystal. It is related to the high ability of GA to adsorb water from the environment, which has been proven in chitosan–gum arabic polyelectrolyte complex film studies. In these examinations, it was proven that the addition of arabic gum to the film formulation causes a greater absorption of water from the environment, which translated into an improvement in mucoadhesive properties [41]. In addition, tragacanth stored in conditions of increased humidity showed the ability of the grains to aggregate into larger clusters (Figure 6f).

In the microscopic image of gum arabic and tragacanth, a slight reduction in the number of grains was observed, especially for tragacanth stored for 120 min under UV access (Figure 6g), and treated thermally (Figure 6h). This fact may be due to the gradually vaporization of adsorbed water under the influence of thermal and UV factors [42].

## 3. Materials and Methods

### 3.1. Tested Samples

In this study, two polymers of natural origin used in pharmacy were tested. Gum arabic (GA) from acacia tree and gum tragacanth (GT) were purchased from Sigma–Aldrich Company (St. Louis, MO, USA).

### 3.2. Ultraviolet Irradiation

Gum arabic and tragacanth were exposed to UVA (315–400 nm) radiation. The irradiation was performed by Medison 250 lamp produced by Schulze & Bohm (Brühl, Germany). The lamp contained 4 radiators with a power of 20 W. Natural polymers in the form of a thin layer of powder were placed in Petri dishes and exposed to UVA radiation from a distance of 30 cm. In the experiment, 30, 60, 90, and 120 min of exposure to ultraviolet irradiation were used.

### 3.3. Higher Temperature

Gum arabic and tragacanth were stored at temperatures of 40, 50, 60, 70, and 80 °C for 60 min. The samples were heated in the professional hot air oven with air circulation of Memmert Company (Schwabach, Germany).

### 3.4. Impact of Relative Humidity, Temperature, and Time on Vapor Sorption

In these studies, the correct drug storage conditions defined by the standards [43], i.e., room temperature 25 °C (±2 °C) and relative humidity (RH) below 60% (±5%) were used. The temperature of 25 °C and RH of 45% were chosen, as these are the values most often found in drug warehouses in Poland [44]. In addition, drug storage parameters defined as abnormal were used. Two types of inappropriate conditions were selected. Proper temperature (25 °C) and elevated RH (75%) and improper temperature (40 °C) and elevated RH (65%).

Natural polymers in the form of a thin layer of powder were placed in Petri dishes and placed in an incubator with constant parameters: 25 °C (±2 °C)/RH 45% (±5%), 25 °C (±2 °C)/RH 75%(±5%), and 40 °C (±2 °C)/RH 65%(±5%)). Relative humidity (RH) values were controlled using a hygrometer produced by Fisher Scientific (Suwanee, GA, USA).

### 3.5. TGA and c-DTA Examination

The thermal stability of gum arabic and tragacanth was determined by thermogravimetric analysis. Thermogravimeter TG 209 F3 Tarsus produced by Netzsch (Selb, Germany) was used. For tested samples thermogravimetric dynamic and isothermal measurements were performed.

For dynamic measurements, the TG, DTG, and c-DTA curves were recorded for 10 mg of tested samples at a heating rate of 10 K/min. in the temperature range of 35–600 °C under N_2_ atmosphere.

For the initial sample of gum arabic and tragacanth, calculated DTA (c-DTA) for endothermal and exothermal effects were performed. In this method the multiple-point temperature calibration was carried out by means of c-DTA. For this purpose, the onset temperatures of the melting peaks of high-purity reference materials (In, Sn, Zn, Al, BaCO_3_, and Au) over the entire temperature range were performed.

For isothermal measurements, the TG curves were recorded for 10 mg of tested samples at a temperature of 50 °C for 60 min under N_2_ atmosphere. Additionally, for initial samples of GA and GT, isothermal measurements in the temperature of range 40 to 80 °C for 60 min were performed.

For dynamic and isothermal analysis, the total flow nitrogen rate was the 50 mL/min. Al_2_O_3_ crucible type used for measurement. The sample curves were analyzed using Proteus 8.0 software produced by Netzsch Company (Selb, Germany).

### 3.6. Colorimetric Analysis

For tested samples, colorimetric analysis in the CIE L*a*b* color system was performed. The NH 310 colorimeter produced by 3nh (China) was used. Analyses of color parameters were done for gum arabic and tragacanth not subjected to and exposed to the tested physical factors. GA and GT were measured 15 min after exposure to all tested storage conditions. All measurements were done three times for each sample. The received values were averaged.

The parameters—lightness (L*), redness (a*), and yellowness (b*)—were used to analyze changes in the color of the tested natural polymers. Parameter L* refers to the lightness of the samples and it ranges from 0 to 100, where 0 is black color and 100 is white color. Parameter a* refers to the redness of the samples. A negative value of a* indicates green, while a* positive one indicates red color. Parameter b* refers to the yellowness of the samples. Positive and negative b* indicate yellow and blue color, respectively [36,45].

Additionally, total color difference (ΔE*) measurements were performed. For this purpose, the formula 1 below was used [36].
ΔE* = [(ΔL*)^2^ + (Δa*)^2^ + (Δb*)^2^]^1/2^(1)
where:

ΔL*—are indicators of difference in lightness between nonexposed and exposed to higher temperature and UV radiation natural polymers;

Δa*—are indicators of difference in redness between nonexposed and exposed to higher temperature and UV radiation natural polymers;

Δb*—are indicators of difference in yellowness between nonexposed and exposed to higher temperature and UV radiation natural polymers.

For tested samples, the browning index (BI) was calculated according formula 2 using the parameters obtained for L*, a*, b* [46]:BI = 100[(x − 0.31)]/0.172(2)
where parameter x was calculated by (3):x = (a* + 1.75 L*)/(5.645 L* + a* − 3.012 b*)(3)

### 3.7. UV Spectrophotometry Analysis

UV spectra of the gum arabic and tragacanth non-exposed and exposed to a tested storage conditions were analyzed in the study. For this purpose, 10 mg of each of the tested polymers were dissolved in 40 mL of water [47]. The samples were then thoroughly mixed and poured into a quartz cuvette with an optical path of 1 cm, which was placed in a spectrophotometer. UV absorbance spectra were recorded in the wavelength range from 200 to 380 nm [47,48,49]. UV-Vis spectrophotometer Thermo Genesys 10S produced by Thermo Scientific (Waltham, MA, USA) was used. Analysis of recorded UV spectra was performed using the programs VisionLite produced by Thermo Scientific (Waltham, MA, USA) and Origin 2016 produced by OriginLab (Northampton, MA, USA).

### 3.8. Optical Microscopy Observations

In the study optical microscope with camera 5MPX produced by Opta-Tech (Warsaw, Poland) was used. The magnification of the recorded image (objective x eyepiece) was 10× and 40× with numerical apertures 0.25 and 0.65, respectively. The gum arabic and tragacanth stored in a closed and open package exposed to tested physical conditions were examined.

### 3.9. Statistical Analysis

The measurements were performed three times for each sample. The results were presented as mean (±) standard deviations (SD). One-way ANOVA test were used to assess statistical significance. The significance level was assumed *p* < 0.05. Statistical analysis was done using the Statistica software produced by TIBCO Software Inc. (Palo Alto, CA, USA).

## 4. Conclusions

The performed examination of influence of different physical factors such as UV-radiation, higher temperature, and relative humidity during storage of gum arabic and tragacanth pointed out that GA is more resistant to higher temperatures compared to GT. Thermogravimetric dynamic analysis (TGA) and c-DTA showed a two-step decomposition both polymers. The first stage was related with the release of absorbed water by GA and GT. The second step was related to the complete decomposition of the tested samples. Moreover, thermogravimetric isothermal analysis (TGA) showed the greatest mass lost for gum tragacanth and gum arabic was recorded at temperatures of 50 and 60 °C, respectively.

The storage temperatures (40–80 °C) used in the experiment indicated no or a barely visible color change in system CIE L*a*b*. This indicates the relative resistance of the tested polymers to the applied temperature conditions. It is additionally confirmed by the recorded UV spectra and the optical microscopy images.

The evaluation of the influence of UV-radiation on GA and GT, analytical methods used in this work, shows the relative stability of the tested polymers. Slightly larger changes were recorded for gum arabic compared to tragacanth, which may indicate potential changes in GA under the influence of UV-radiation.

The largest assessed changes occurred in GA and GT under the influence of storage conditions in high relative humidity. Polymers stored in high relative humidity (75% RH) showed the highest weight loss in TGA isothermal measurements. It proves high water sorption by the samples under these conditions. Colorimetric analysis showed a significant color change in the system CIE L*a*b* of polymers exposed to high relative humidity (75%RH/25 °C) and high relative humidity and temperature (65%RH/40 °C) in relation to the initial samples. This indicates changes occurring in the tested polymers as a result of their exposure to the tested physical factors. The most visible changes in the optical microscopy images were visible for samples stored under the conditions of 75% RH/25 °C/3 days. Especially this observations concerned for gum arabic.

UV absorbance spectra for GA showed a hypochromic effect for all storage conditions. In turn, only GT stored under conditions 75% RH/25 °C/3 days showed a hyperchromic effect.

## Figures and Tables

**Figure 1 molecules-27-01510-f001:**
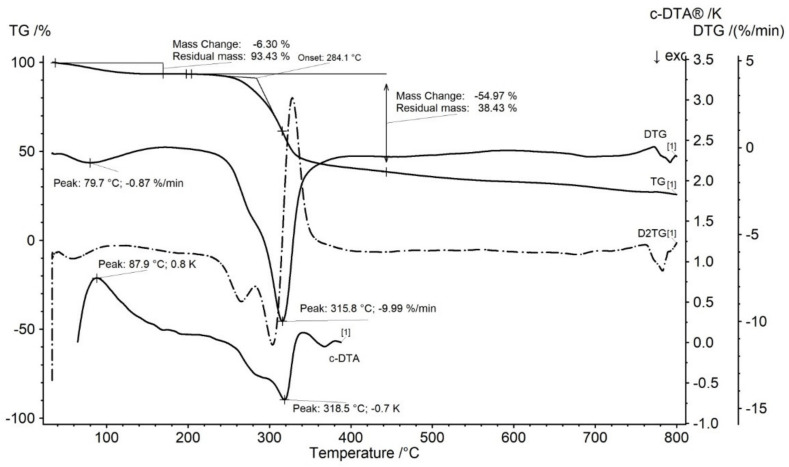
TG, DTG, D2TG, and c-DTA curves of gum arabic (GA). Curves recorded for dynamic thermogravimetric measurement.

**Figure 2 molecules-27-01510-f002:**
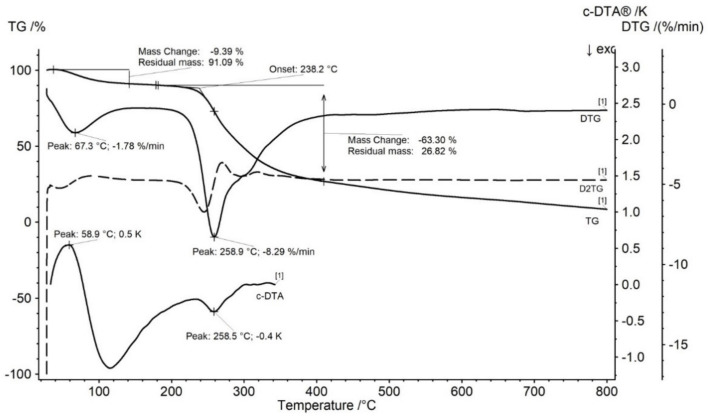
TG, DTG, D2TG, and c-DTA curves of gum tragacanth (GT). Curves recorded for dynamic thermogravimetric measurement.

**Figure 3 molecules-27-01510-f003:**
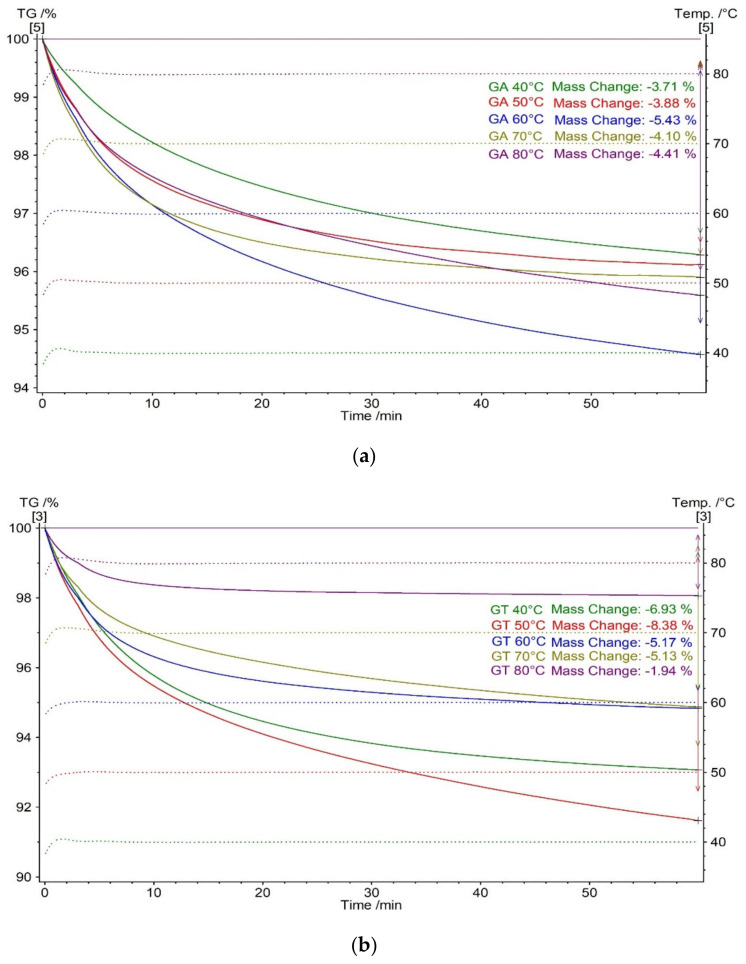
TG isothermal curves of (**a**) gum arabic and (**b**) gum tragacanth heated 60 min in the range of 40–80 °C.

**Figure 4 molecules-27-01510-f004:**
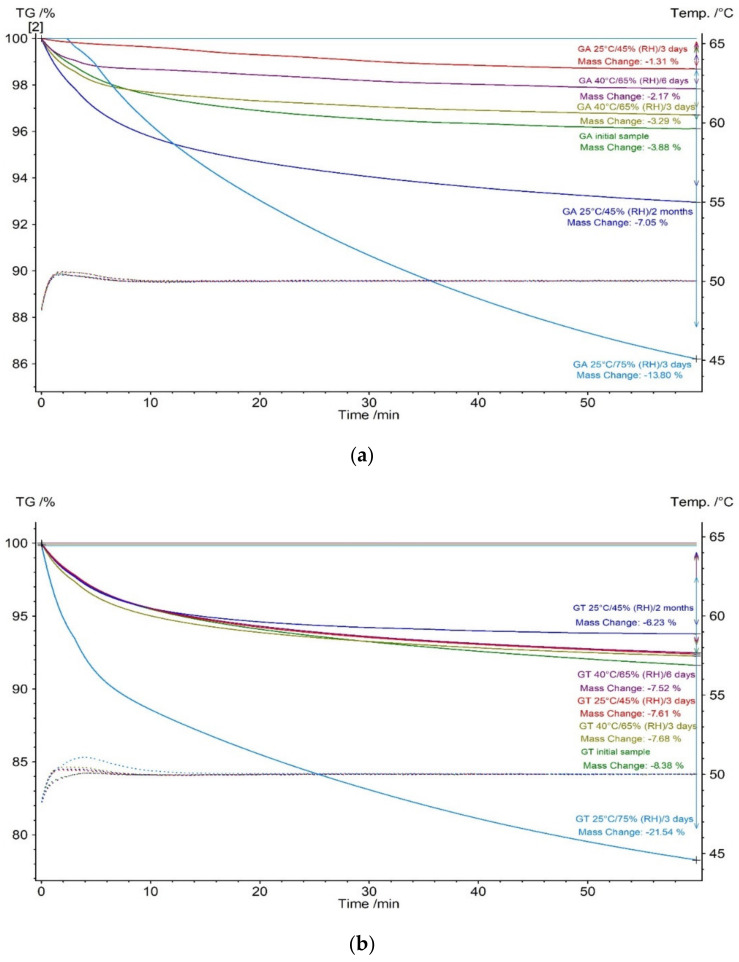
TG isothermal curves of (**a**) gum arabic and (**b**) gum tragacanth storage in different conditions (temperature/RH/time). Measurements were performed at 50 °C for 60 min.

**Figure 5 molecules-27-01510-f005:**
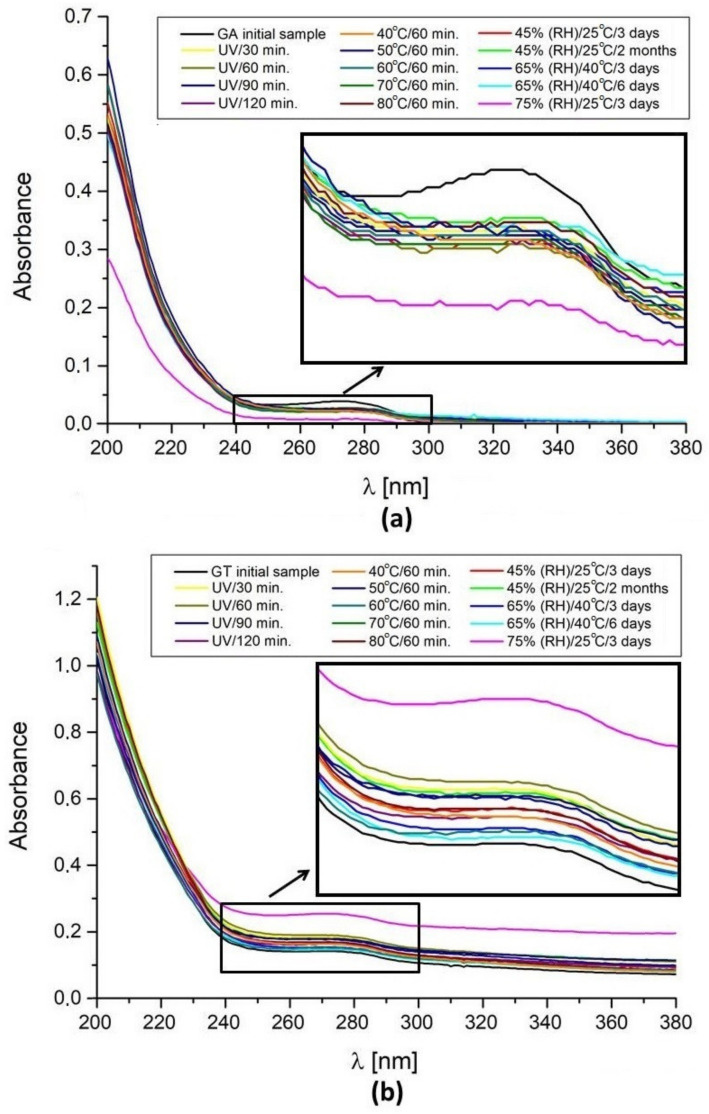
UV spectra of (**a**) gum arabic, and (**b**) gum tragacanth in the tested storage conditions. Measurements was performed at room temperature in the wavelength range 200–380 nm.

**Figure 6 molecules-27-01510-f006:**
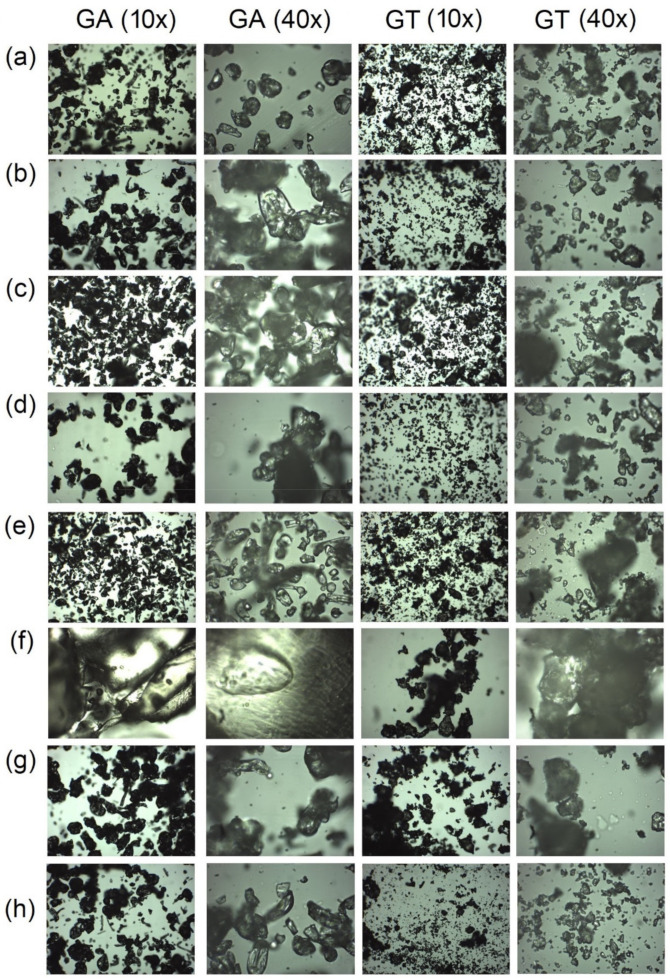
Optical microscopy images of (**a**) gum arabic (GA) and gum tragacanth (GT) stored in a closed package (initial samples). GA and GT stored in room conditions (25 °C, 45%RH) in open package for (**b**) 3 days and (**c**) 2 months. GA and GT stored in under conditions of accelerated aging (40 °C, 65%RH) in the open package for (**d**) 3 and (**e**) 6 days. (**f**) GA and GT stored in room temperature, and in increased relative humidity (25 °C, 75%RH) in open package for 3 days. (**g**) GA and GT UV-irradiated during 120 min. (**h**) GA and GT storage in temperature 80 °C for 60 min.

**Table 1 molecules-27-01510-t001:** Analysis of color in the CIE L*a*b* space, total color difference (ΔE), and browning index (BI) parameters for gum arabic storage in different conditions.

Storage Conditions		L* [±SD]	a* [±SD]	b* [±SD]	ΔE [±SD]	BI [±SD]
Initial sample		82.44 [±0.01]	10.08 [±0.01]	42.65 [±0.03]	-	79.31 [±0.02]
UV irradiation	UV/30 min	82.22 [±0.01]	9.71 [±0.03]	42.27 [±0.01]	0.58 [±0.02]	78.37 [±0.02]
	UV/60 min	81.05 [±0.04]	10.48 [±0.01]	42.91 [±0.01]	1.47 [±0.02]	82.12 [±0.02]
	UV/90 min	81.42 [±0.05]	9.52 [±0.03]	41.35 [±0.04]	1.74 [±0.04]	77.12 [±0.04]
	UV/120 min	82.19 [±0.01]	9.77 [±0.02]	42.10 [±0.01]	0.68 [±0.01]	78.08 [±0.01]
Temperature	40 °C/60 min	83.55 [±0.04]	10.42 [±0.01]	42.51 [±0.03]	1.17 [±0.03]	77.02 [±0.03]
	50 °C/60 min	82.23 [±0.03]	10.37 [±0.02]	43.33 [±0.04]	0.76 [±0.03]	81.41 [±0.03]
	60 °C/60 min	82.66 [±0.05]	10.34 [±0.03]	39.44 [±0.02]	0.35 [±0.03]	79.11 [±0.03]
	70 °C/60 min	81.69 [±0.05]	10.40 [±0.03]	42.64 [±0.04]	0.82 [±0.04]	80.57 [±0.04]
	80 °C/60 min	81.43 [±0.05]	11.01 [±0.03]	42.58 [±0.04]	1.37 [±0.04]	81.36 [±0.04]
Relative humidity/temperature/ time	45% (RH)/25 °C/3 days	82.53 [±0.02]	10.05 [±0.01]	43.21 [±0.04]	0.57 [±0.02]	80.44 [±0.02]
	45% (RH)/25 °C/2 months	81.76 [±0.03]	10.35 [±0.03]	42.08 [±0.03]	0.93 [±0.03]	79.14 [±0.03]
	65% (RH)/40 °C/6 days	70.66 [±0.05]	11.71 [±0.02]	38.80 [±0.03]	12.50 [±0.03]	88.74 [±0.03]
	75% (RH)/25 °C/3 days	64.65 [±0.02]	9.38 [±0.01]	34.56 [±0.04]	19.55 [±0.02]	84.33 [±0.02]

The results are presented as mean ± SD; The results were considered statistically significant when *p* < 0.05.

**Table 2 molecules-27-01510-t002:** Analysis of color in the CIE L*a*b* space, total color difference (ΔE), and browning index (BI) parameters for gum tragacanth storage in different conditions.

Storage Conditions		L* [±SD]	a* [±SD]	b* [±SD]	ΔE [±SD]	BI [±SD]
Initial sample		80.33 [±0.02]	11.39 [±0.02]	39.89 [±0.01]	-	76.98 [±0.02]
UV irradiation	UV/30 min	78.60 [±0.02]	10.49 [±0.02]	38.63 [±0.04]	2.32 [±0.03]	75.42 [±0.03]
	UV/60 min	80.35 [±0.02]	11.41 [±0.03]	39.76 [±0.04]	0.14 [±0.03]	76.68 [±0.03]
	UV/90 min	79.30 [±0.03]	11.15 [±0.01]	39.37 [±0.04]	1.18 [±0.03]	76.87 [±0.03]
	UV/120 min	80.13 [±0.02]	11.33 [±0.03]	39.75 [±0.01]	0.25 [±0.02]	76.86 [±0.02]
Temperature	40 °C/60 min	80.70 [±0.05]	11.54 [±0.03]	40.01 [±0.04]	0.42 [±0.04]	76.92 [±0.04]
	50 °C/60 min	79.84 [±0.05]	11.63 [±0.02]	39.91 [±0.01]	0.55 [±0.03]	77.89 [±0.03]
	60 °C/60 min	80.96 [±0.04]	11.44 [±0.03]	39.44 [±0.02]	0.77 [±0.03]	75.24 [±0.03]
	70 °C/60 min	80.10 [±0.03]	11.65 [±0.02]	40.04 [±0.01]	0.38 [±0.02]	76.86 [±0.02]
	80 °C/60 min	79.93 [±0.05]	11.73 [±0.02]	39.78 [±0.04]	0.54 [±0.04]	77.56 [±0.04]
Relative humidity/temperature/ time	45% (RH)/25 °C/3 days	80.75 [±0.02]	11.32 [±0.04]	39.85 [±0.01]	0.42 [±0.02]	76.30 [±0.02]
	45% (RH)/25 °C/2 months	80.27 [±0.04]	11.44 [±0.03]	39.53 [±0.02]	0.37 [±0.03]	76.29 [±0.03]
	65% (RH)/40 °C/6 days	72.78 [±0.05]	11.38 [±0.01]	36.37 [±0.04]	8.33 [±0.03]	78.67 [±0.03]
	75% (RH)/25 °C/3 days	69.83 [±0.01]	12.85 [±0.03]	39.82 [±0.02]	10.60 [±0.02]	94.35 [±0.02]

The results are presented as mean ± SD. The results were considered statistically significant when *p* < 0.05.

## Data Availability

Data available on request from authors. The data that support the findings of this study are available from the corresponding author upon reasonable request.

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
