# Peer review of "Influence of Storage Conditions on the Stability of Gum Arabic and Tragacanth"

_molecules, 2022, doi:10.3390/molecules27051510_

Round 1
Reviewer 1 Report
keywords should not be limited only to listing the research material and the methods used - suggests correcting, they do not facilitate the search for the topic of work
line 32-35 - too generally described influence of air / ambient humidity on API durability
fig 3c i 4c harmonize graphically with figs 3a/4a and 3b/4b
Table 1 and 2 line "Storage conditions" L * [± 0.05] ??? what is reading is this a standard deviation? Such presentation of the results raises doubts about the reliable measurement or their conversion. Even if for each of the parameters L * a * b there is the same value of the standard deviation ... which I strongly doubt, it should be taken into account for each of the parameters.
Fig 5 describing A should start with 0 - negative values indicate analytical errors, although the fact of the appearance of a value below zero but close may also be the issue of the sensitivity of the spectrophotometer ...
it is necessary to correct the section "3.4. Impact of relative humidity, temperature, and time on water vapor sorption", in principle, two concepts of humidity levels are doubled - vapor sorption in water. In addition, there is no description of how the relative air humidity is controlled. There are also explanations why the measurements were made in a strange and, as for me, randomly selected temperature of 25 degrees Celcius for 45% RH and 75% at 40 degrees C for 65% RH - no consistency.
3.7. Analysis of UV-Vis spectrophotometry - why does UV-Vis appear in the title if the wavelength (nm) 200-380 is UV?
applications should be written in continuous text, without bullet points 1-6,
line 340 - UV-VIs ???
Author Response
REVIEWER 1
Comments and Suggestions, and Answers
We are very grateful for the valuable opinion and comments. All the suggested valuable changes have done in the corrected version of our paper.
1) ,,keywords should not be limited only to listing the research material and the methods used - suggests correcting, they do not facilitate the search for the topic of work’’
Answer for Reviewer
As suggested by the Reviewer the keywords was change.
New version:
gum arabic; tragacanth; proper storage conditions; UV-irradiation; higher temperature; relative humidity; TGA; c-DTA; colorimetric analysis; UV spectrophotometry; optical microscope observations
2) ,,line 32-35 - too generally described influence of air / ambient humidity on API durability’’
Answer for Reviewer
As suggested by the Reviewer in the Introduction was extended described influence of air / ambient humidity on API durability.
,,The active pharmaceutical ingredient or excipients may be decomposed by the hydrolysis process by the action of steam or water [10]. The derivatives of carboxylic acids (esters, lactones, lactams, amides) show the greatest sensitivity to this type of reaction [11]. Moisture can also influence on crystalline system of APIs [12,13]. This applies especially to unstable or metastable amorphous forms which, under the influence of increased humidity, tend to change the internal order consisting in reaching the form with the lowest internal energy [12]. This may alter the release profile of the drug or influence on bioavailability of the drug substance [11, 13]. Increased moisture sorption by the finished drug formulation may cause an increase in microbial contamination [14]. Also substances sensitive to increased humidity or light can be decomposed by oxidation [15]. Particularly sensitive to oxidation reactions are thiols, thioethers, ethers, aldehydes, but also natural gums [11]. Oxidation processes are often accompanied by a change in color or odor, which makes them easier to observe [11].’’
[10] Tomar, M.; Singh, A.K.; Sinha, A.R. Effect of moisture content of excipient (microcrystalline cellulose) on direct compressible solid dosage forms. Int. J. Pharm. Sci. Res. 2017, 8(1), 282-288.
[11] Sznitowska, M (Ed.). Farmacja Stosowana; PZWL: Warsaw, Poland, 2017; pp. 795, 798, 896, 925.
[12] Prudic, A.; Ji, Y.; Luebbert, C.; Sadowski, G. Influence of humidity on the phase behavior of API/polymer formulations. Eur. J. Pharm. Biopharm. 2015, 94, 352-362.
[13] Rumondor, A.C.F.; Stanford, L.A.; Taylor, L.S. Effects of polymer type and storage relative humidity on the kinetics of felodipine crystalilization from amorphous solid dispersions. Pharm. Res. 2009, 26(12), 2599-2606.
[14] Tomar, M.; Singh, A.K.; Sinha, A.R. Power and tablet profile of microcrystalline cellulose (MCC) of different with degree of polymerization. Int. J. Rec. Sci. Res. 2016, 7(6), 12044-12047.
[15] Hovorka, S.W.; Schoneich C. Oxidative degradation of pharmaceuticals: theory, mechanisms and inhibition. J. Pharm. Sci. 2001, 90(3), 253-269.
3) ,,fig 3c i 4c harmonize graphically with figs 3a/4a and 3b/4b’’
Answer for Reviewer
As suggested by the Reviewer the figures 3c, and 4c was deleted.
4) ,,Table 1 and 2 line "Storage conditions" L * [± 0.05] ??? what is reading is this a standard deviation? Such presentation of the results raises doubts about the reliable measurement or their conversion. Even if for each of the parameters L * a * b there is the same value of the standard deviation ... which I strongly doubt, it should be taken into account for each of the parameters.’’
Answer for Reviewer
The results are presented as standard deviation (±SD). For each parameter, the maximum standard deviation obtained for all results was originally given. However, as indicated by the Reviewer, a change was made and the standard deviation was given for each parameter separately.
5) ,,Fig 5 describing A should start with 0 - negative values indicate analytical errors, although the fact of the appearance of a value below zero but close may also be the issue of the sensitivity of the spectrophotometer ...’’
Answer for Reviewer
As indicated by the Reviewer, the figure 5 has been corrected.
6) ,,it is necessary to correct the section "3.4. Impact of relative humidity, temperature, and time on water vapor sorption", in principle, two concepts of humidity levels are doubled - vapor sorption in water. In addition, there is no description of how the relative air humidity is controlled. There are also explanations why the measurements were made in a strange and, as for me, randomly selected temperature of 25 degrees Celcius for 45% RH and 75% at 40 degrees C for 65% RH - no consistency.’’
Answer for Reviewer
As indicated by the Reviewer, the section 3.4. has been corrected.
,, 3.4. Impact of relative humidity, temperature, and time on vapor sorption
In this studies the correct drug storage conditions defined by the standards [43], i.e. room temperature 25°C (±2°C) and relative humidity (RH) below 60% (±5%) were used. The temperature of 25°C, and RH of 45% were chosen, as these are the values most often found in drug warehouses in Poland [44]. In addition, drug storage parameters defined as abnormal were used. Two types of inappropriate conditions have been selected. Proper temperature (25°C) and elevated RH (75%) and improper temperature (40°C) and elevated RH (65%).
Natural polymers in the form of a thin layer of powder were placed in Petri dishes and placed in an incubator with constant parameters: 25°C (±2°C)/RH 45% (±5%), 25°C (±2°C)/RH 75% (±5%), and 40°C (±2°C)/RH 65% (±5%). Relative humidity (RH) values were controlled using a hygrometer produced by Fisher Scientific (Suwanee, Ga, USA).’’
[43] Guide to good storage practices for pharmaceuticals Annex 9, WHO Technical Report Series, No. 908, World Health Organization, 2003
[44] Kucharczyk, M.; Slusarski, B. Mapowanie rozkładu temperatury w przestrzeniach magazynowych – praktyczne podejście. Farm. Pol. 2009, 65(10), 707-712.
In addition, other researchers [1-4] testing polymers used in their works, inter alia, applied temperature conditions (25°C, 40°C), and RH (45%, 65%, 75%) in different compilations.
[1] Rumondor, A.C.F.; Stanford, L.A.; Taylor, L.S. Effects of polymer type and storage relative humidity on the kinetics of felodipine crystalilization from amorphous solid dispersions. Pharm. Res. 2009, 26(12), 2599-2606.
[2] Rumondor, A.C.F.; Taylor, L.S. Effect of polymer hygroscopicity on the phase behavior of amorphous solid dispersions in the presence of moisture. Molecular Pharmaceutics 2009, 7(2), 477-490.
[3] Collier, J.W.; Shah, R.B.; Gupta, A.; Sayeed, V.; Habib, M.J.; Khan, M.A. Influence of formulation and processing factors on stability of levothyroxine sodium pentahydrate. Pharm. Sci. Tech. 2010, 11(2), 818-825.
[4] Marsac, P.J.; Konno, H.; Rumondor C.F.R.; Taylor, L.S. Recrystallization of nifedypine and felodipine from amorphous molecular level solid dispersions containing poly(vinylpyrrolidone) and sorbed water. Pharm. Res. 2008, 25(3), 647-656..
7) ,,3.7. Analysis of UV-Vis spectrophotometry - why does UV-Vis appear in the title if the wavelength (nm) 200-380 is UV?’’
Answer for Reviewer
As indicated by the Reviewer, the title of chapter 3.7. has been corrected.
8) ,,applications should be written in continuous text, without bullet points 1-6,’’
Answer for Reviewer
As indicated by the Reviewer, the Conclusions were reorganized.
,,The performed examination of influence of different physical factors like UV-radiation, higher temperature, and relative humidity during storage of gum arabic and tragacanth pointed out that GA is more resistant to higher temperatures compared to GT. Thermogravimetric dynamic analysis (TGA) and c-DTA showed a two-step decomposition both polimers. The first stage is related with the release of absorbed water by GA and GT. The second step is related to the complete decomposition of the tested samples. Moreover, thermogravimetric isothermal analysis (TGA) showed the greatest mass lost for gum tragacanth, and gum arabic were recorded at temperature 50°C and 60°C respectively.
The storage temperatures (40°C-80°C) used in experiment indicated no or a barely visible color change in system CIE L*a*b* . This indicates the relative resistance of the tested polymers to the applied temperature conditions. It is additionally confirmed by the recorded UV spectra and the optical microscopy images.
The evaluation of the influence of UV-radiation on GA and GT, analytical methods used at work, shows the relative stability of the tested polymers. Slightly larger changes were recorded for gum arabic compared to tragacanth. Which may indicate potential changes in GA under the influence of UV-radiation.
The largest assessed changes occurred in GA and GT under the influence of storage conditions in high relative humidity. Polymers stored in high relative humidity (75% RH) showed the highest weight loss in TGA isothermal measurements. It proves high water sorption by the samples under these conditions. Colorimetric analysis showed a significant color change in the system CIE L*a*b* of polymers exposed to high relative humidity (75%RH/25°C) and high relative humidity and temperature (65%RH/40°C) in relation to the initial samples. This indicates changes occurring in the tested polymers as a result of their exposure to the tested physical factors. The most visible changes in the optical microscopy images were visible for the samples stored under the conditions of 75% RH/25°C/3 days. Especially this observations concerned for gum arabic.
UV absorbance spectra for GA showed a hypochromic effect for all storage conditions. In turn, only GT stored under conditions 75% RH/25°C/3 days showed a hyperchromic effect.’’
9) ,,line 340 - UV-VIs ???’’
Answer for Reviewer
As indicated by the Reviewer, the UV-Vis in line 340 has been corrected.

Reviewer 2 Report
Authors have investigated the influence of various storage conditions on the stability of gum Arabic and tragacanth. The influence of various stress conditions on the stability of gum Arabic and tragacanth was investigated using various qualitative techniques, including TG, c-DTA, colorimetric analysis, uv-visible spectrometry, and optical microscopy. Colorimetric and uv-visible spectrometry techniques are not recommended for the stability evaluations. Authors have evaluated only physical stability. The main focus of manuscript is the stability evaluation, and hence chemical decomposition of gum Arabic and tragacanth must be evaluated using highly sensitive stability-indicating techniques. The level of English language is too poor. The quality of all graphics is also poor. These are routine qualitative studies, which are not suitable for publication in reputed journals like Molecules.
Author Response
REVIEWER 2
Comments and Suggestions, and Answers
,,Authors have investigated the influence of various storage conditions on the stability of gum Arabic and tragacanth. The influence of various stress conditions on the stability of gum Arabic and tragacanth was investigated using various qualitative techniques, including TG, c-DTA, colorimetric analysis, uv-visible spectrometry, and optical microscopy. Colorimetric and uv-visible spectrometry techniques are not recommended for the stability evaluations. Authors have evaluated only physical stability. The main focus of manuscript is the stability evaluation, and hence chemical decomposition of gum Arabic and tragacanth must be evaluated using highly sensitive stability-indicating techniques. The level of English language is too poor. The quality of all graphics is also poor. These are routine qualitative studies, which are not suitable for publication in reputed journals like Molecules.’’
Answer for Reviewer
We are grateful for the opinion and comments.
The work uses routine techniques available to most people. It has been assessed whether basic techniques such as colorimetric analysis, UV-spectrophotometry, and optical microscopy can be used for screening assessment of storage conditions. Additionally, a more advanced technique such as TGA and c-DTA was used. Often a change in color is the first symptom indicating changes in the substance under the influence of physical factors. We agree that the mechanism of thermal decomposition of the compounds requires further research. We would like to continue our studies of thermal decomposition of the substances by the use of thermogravimetric method and FT-IR.
There are many scientific papers that use colorimetric analysis to evaluate the effects of physical conditions on drugs and food. Below are some examples of the use of colorimetric analysis:
Echavarria ,A.P.; Pagan, J.; Ibarz, A. Kinetics of color development in glucose/Amino Acid model systems at different tem-peratures. Sci. Agropec. 2016, 7(1), 15-21.
Singh,B.; Parwate, D.V.; Shukla, S.K. Radiosterilization of fluoroquinolones and cephalosporins: assessment of radiation damage on antibiotics by changes in optical property and colorimetric parameters. AAPS Pharm. Sci. Tech. 2009, 10(1), 34-43.
Kuan, Y.H.; Bhat, R.; Senan, C.; Williams, P.A.; Karim, A.A. Effects of ultraviolet on the physicochemical and functional properties of gum Arabic. J. Agric. Food Chem. 2009, 57, 9154-9159.
Magdalena Tworek, M.; Skowroński, Ł.; Makarewicz, E.; Kowalik, J. Properties of poly (vinyl chloride) membranes con-taining cadmium pigments, irradiated with UV irradiation. Sci. Rep. 2021, 11(1816), 1-11.
Anna Masek, A.; Latos-Brozio, M. The effect of substances of plant origin on the thermal and thermo-oxidative ageing of aliphatic polyesters (PLA, PHA). Polymers 2018, 10,(1252), 1-17.
Popa, S.; Boran, S.; Davidescu, C. Influence of storage conditions upon CIELAB and thermal properties of sesame food oil. Rev .Chim. (Bucharest) 2016, 67(7), 1290-1293.
Zhoua, R.; Sub, S.; Yan, L.; Li, Y. Effect of transport vibration levels on mechanical damage and physiological responses of Huanghua pears (Pyrus pyrifolia Nakai, cv. Huanghua). Posth. Biol. and Tech. 2007, 46, 20–28.
Pathare, P.B.; Opara, U.L.; Al-Julanda Al-Said, F. Colour measurement and analysis in fresh and processed foods: a review. Food Biop. Tech. 2013, 6, 36–60.
There is little research in the scientific literature on the analysis of natural polymers using basic research techniques. The research performed by us can be helpful and continued by the others analytical technics. In the future we would like to continue our studies of influence of storage conditions on the gum arabic and tragacanth by the use of EPR and FT-IR method.
The new version of our manuscript, after the revision according to the Reviewers’ comments, was corrected by an English translator. The cases of spelling were corrected. The quality of figures was corrected.

Reviewer 3 Report
Minor remarks
- All minor remarks are highlighted in the manuscript.
Major remarks
- Each reference is desirable to be discussed separately. Avoid lumping the references in the manuscript.
- The obtained results should be statistically processed (for example, the reduction should be expressed as a percentage) and compared with the literature data for this or a similar polymer.

Author Response
REVIEWER 3
Comments and Suggestions, and Answers
We are very grateful for the valuable opinion and comments. All the suggested valuable changes have done in the corrected version of our paper.
1) Minor remarks
,,All minor remarks are highlighted in the manuscript.’’
Answer for Reviewer
All minor remarks marked in the text by the Reviewer have been corrected.
2) Major remarks
,,Each reference is desirable to be discussed separately. Avoid lumping the references in the manuscript.’’
Answer for Reviewer
We are very grateful the Reviewer for valuable attention. The comment has been included in the text where it was possible.
,,The obtained results should be statistically processed (for example, the reduction should be expressed as a percentage) and compared with the literature data for this or a similar polymer.’’
Answer for Reviewer
Information on statistical analysis has been added in section ,,Materials and Methods’’, point 3.9.
,,3.9. Statistical analysis
The measurements was performed three times for each sample. The results are presented as mean (±) standard deviations (SD). One-way ANOVA test were used to assess statistical significance. The significance level was assumed p<0.05. Statisitcal analysis was done using the Statistica software produced by TIBCO Software Inc. (Palo Alto, CA, USA).’’
The work discussion has been extended:
,,Similar results were observed by Zografi G. et al. in the poly(vinylpyrrolidone) studies [30]. They recorded less water sorption for the polymer stored at a higher temperature [30]. This fact good corresponding also with other measurements [31].’’
[30] Oksanen, C.A.; Zografi, G. The relationship beetween the glass transition temperature and water vapor absorption by poly(vinylpyrrolidone). Pharm. Res. 1990, 7(6), 654-657.
,,In another work, poly (vinyl chloride) and PVC membranes containing cadmium pig-ments was exposed to UV irradiation [35]. Minimal changes in the color of tested synthetic polymer was observed [35].’’
[35] Magdalena Tworek, M.; Skowroński, Ł.; Makarewicz, E.; Kowalik, J. Properties of poly (vinyl chloride) membranes containing cadmium pigments, irradiated with UV irradiation. Sci. Rep. 2021, 11(1816), 1-11.
,,A minimal change in total color difference (DE*) parameter for polylactide (PLA) treated with 100°C was recorded by other researchers [37]. The same researchers registered a large change in the DE* parameter for polyhydroxyalkanoate (PHA), which indicates the high sensitivity of this polymer to exposure on higher temperature [37].’’
[37] Anna Masek, A.; Latos-Brozio, M. The effect of substances of plant origin on the thermal and thermo-oxidative ageing of aliphatic polyesters (PLA, PHA). Polymers 2018, 10,(1252), 1-17.
,,For gum arabic storage at higher temperature (40°C) with increased relative humidity compared to room temperature (25°C) with normal, and increased relative humidity caused a decrease in the amount of water in the sample. Similar results for Acacia gum were reported in another work [32]. They was recorded for acacia gum a lower content of water with increasing storage temperature [32]. Mansoori N. et al. explained this phenomen that the higher temperatures, intermolecular forces of attraction diminish because of the increased kinetic energy of molecules. This resulting in increased molecular mobility. For that reason, water molecules, which have non limited motions, are expected to form weaker bonds with the binding sites of polymer molecules [32].’’
[32] Mansoori, N.; Majzoobi, M.; Gavahian, M.; Badii, F.; Farahnaky, A. Acacia gum as a natural anti-plasticizer for the production of date syrup powder: sorption isotherms, physicochemical properties, and data modeling. Foods 2020, 9(50), 1-17.

Reviewer 4 Report
- The Abstract should be written more quantitative.
- Some related research works should be added and explained in the Introduction section.
- The comparison of results in sections with other research articles should be considered.
- The main goal of this article should be explained more clearly in the Conclusion section.
- The references should be checked in terms of the style and cohesion.
Author Response
REVIEWER 4
Comments and Suggestions, and Answers
We are very grateful for the valuable opinion and comments. All the suggested valuable changes have done in the corrected version of our paper.
1) ,,The Abstract should be written more quantitative.’’
Answer for Reviewer
As indicated by the Reviewer, the Abstract were reorganized.
,,Storage conditions should be so chosen that they do not affects the action and stability of the active pharmaceutical substance (API), and excipients used in pharmacy. UV irrradiation, increased temperature and relative humidity can decompose storage substances by photolysis, thermolysis, and hydrolysis process, respectively. The effect of physical factors may be the decomposition of pharmaceutical substances or their inappropriate action, including pharmacological effects. Polymers of natural origin are increasingly used in the pharmaceutical industry. With this in mind, was to evaluate the effect of storage conditions on the stability of gum arabic (GA), and tragacanth (GT). The influence of higher temperature, UV irradiation, and relative humidity on GA, and GT was tested. Thermogravimetry (TG, c-DTA), colorimetric analysis, UV-Vis spectrophotometry, and optical microscopy were used as research methods. The TGA and c-DTA examination indicated that decomposition of GA is starts at a higher temperature compared to GT. This indicate that gum arabic is more resistant to higher temperatures compared to tragacanth. However, the conducted analyzes showed that gum arabic is more sensitive to the tested storage conditions. Among the tested physical conditions, both polymers were most sensitive to conditions of increased relative humidity in the environment.’’
2) ,,Some related research works should be added and explained in the Introduction section.’’
Answer for Reviewer
As suggested by the Reviewer related research works was added in the Introduction section.
[10] Tomar, M.; Singh, A.K.; Sinha, A.R. Effect of moisture content of excipient (microcrystalline cellulose) on direct compressible solid dosage forms. Int. J. Pharm. Sci. Res. 2017, 8(1), 282-288.
[11] Sznitowska, M (Ed.). Farmacja Stosowana; PZWL: Warsaw, Poland, 2017; pp. 795, 798, 896, 925.
[12] Prudic, A.; Ji, Y.; Luebbert, C.; Sadowski, G. Influence of humidity on the phase behavior of API/polymer formulations. Eur. J. Pharm. Biopharm. 2015, 94, 352-362.
[13] Rumondor, A.C.F.; Stanford, L.A.; Taylor, L.S. Effects of polymer type and storage relative humidity on the kinetics of felodipine crystalilization from amorphous solid dispersions. Pharm. Res. 2009, 26(12), 2599-2606.
[14] Tomar, M.; Singh, A.K.; Sinha, A.R. Power and tablet profile of microcrystalline cellulose (MCC) of different with degree of polymerization. Int. J. Rec. Sci. Res. 2016, 7(6), 12044-12047.
[15] Hovorka, S.W.; Schoneich C. Oxidative degradation of pharmaceuticals: theory, mechanisms and inhibition. J. Pharm. Sci. 2001, 90(3), 253-269.
Especially the description of moisture effects and light on API has been extended with new literature.
,,The active pharmaceutical ingredient or excipients may be decomposed by the hydrolysis process by the action of steam or water [10]. The derivatives of carboxylic acids (esters, lactones, lactams, amides) show the greatest sensitivity to this type of reaction [11]. Moisture can also influence on crystalline system of APIs [12,13]. This applies especially to unstable or metastable amorphous forms which, under the influence of increased humidity, tend to change the internal order consisting in reaching the form with the lowest internal energy [12]. This may alter the release profile of the drug or influence on bioavailability of the drug substance [13] [11]. Increased moisture sorption by the finished drug formulation may cause an increase in microbial contamination [14]. Also substances sensitive to increased humidity or light can be decomposed by oxidation [15]. Particularly sensitive to oxidation reactions are thiols, thioethers, ethers, aldehydes, but also natural gums [11]. Oxidation processes are often accompanied by a change in color or odor, which makes them easier to observe [11].’’
3) ,,The comparison of results in sections with other research articles should be considered.’’
Answer for Reviewer
As indicated by the Reviewer, the discussion of the results was extended.
,,Similar results were observed by Zografi G. et al. in the poly(vinylpyrrolidone) studies [30]. They recorded less water sorption for the polymer stored at a higher temperature [30]. This fact good corresponding also with other measurements [31].’’
[30] Oksanen, C.A.; Zografi, G. The relationship beetween the glass transition temperature and water vapor absorption by poly(vinylpyrrolidone). Pharm. Res. 1990, 7(6), 654-657.
,,In another work, poly (vinyl chloride) and PVC membranes containing cadmium pig-ments was exposed to UV irradiation [35]. Minimal changes in the color of tested synthetic polymer was observed [35].’’
[35] Magdalena Tworek, M.; Skowroński, Ł.; Makarewicz, E.; Kowalik, J. Properties of poly (vinyl chloride) membranes containing cadmium pigments, irradiated with UV irradiation. Sci. Rep. 2021, 11(1816), 1-11.
,,A minimal change in total color difference (DE*) parameter for polylactide (PLA) treated with 100°C was recorded by other researchers [37]. The same researchers registered a large change in the DE* parameter for polyhydroxyalkanoate (PHA), which indicates the high sensitivity of this polymer to exposure on higher temperature [37].’’
[37] Anna Masek, A.; Latos-Brozio, M. The effect of substances of plant origin on the thermal and thermo-oxidative ageing of aliphatic polyesters (PLA, PHA). Polymers 2018, 10,(1252), 1-17.
,,For gum arabic storage at higher temperature (40°C) with increased relative humidity compared to room temperature (25°C) with normal, and increased relative humidity caused a decrease in the amount of water in the sample. Similar results for Acacia gum were reported in another work [32]. They was recorded for acacia gum a lower content of water with increasing storage temperature [32]. Mansoori N. et al. explained this phenomen that the higher temperatures, intermolecular forces of attraction diminish because of the increased kinetic energy of molecules. This resulting in increased molecular mobility. For that reason, water molecules, which have non limited motions, are expected to form weaker bonds with the binding sites of polymer molecules [32].’’
[32] Mansoori, N.; Majzoobi, M.; Gavahian, M.; Badii, F.; Farahnaky, A. Acacia gum as a natural anti-plasticizer for the production of date syrup powder: sorption isotherms, physicochemical properties, and data modeling. Foods 2020, 9(50), 1-17.
4) ,,The main goal of this article should be explained more clearly in the Conclusion section.’’
Answer for Reviewer
As indicated by the Reviewer, the main goal was explained more clearly.
,,The performed examination of influence of different physical factors like UV-radiation, higher temperature, and relative humidity during storage of gum arabic and tragacanth pointed out that GA is more resistant to higher temperatures compared to GT. Thermogravimetric dynamic analysis (TGA) and c-DTA showed a two-step decomposition both polimers. The first stage is related with the release of absorbed water by GA and GT. The second step is related to the complete decomposition of the tested samples. Moreover, thermogravimetric isothermal analysis (TGA) showed the greatest mass lost for gum tragacanth, and gum arabic were recorded at temperature 50°C and 60°C respectively.
The storage temperatures (40°C-80°C) used in experiment indicated no or a barely visible color change in system CIE L*a*b* . This indicates the relative resistance of the tested polymers to the applied temperature conditions. It is additionally confirmed by the recorded UV spectra and the optical microscopy images.
The evaluation of the influence of UV-radiation on GA and GT, analytical methods used at work, shows the relative stability of the tested polymers. Slightly larger changes were recorded for gum arabic compared to tragacanth. Which may indicate potential changes in GA under the influence of UV-radiation.
The largest assessed changes occurred in GA and GT under the influence of storage conditions in high relative humidity. Polymers stored in high relative humidity (75% RH) showed the highest weight loss in TGA isothermal measurements. It proves high water sorption by the samples under these conditions. Colorimetric analysis showed a significant color change in the system CIE L*a*b* of polymers exposed to high relative humidity (75%RH/25°C) and high relative humidity and temperature (65%RH/40°C) in relation to the initial samples. This indicates changes occurring in the tested polymers as a result of their exposure to the tested physical factors. The most visible changes in the optical microscopy images were visible for the samples stored under the conditions of 75% RH/25°C/3 days. Especially this observations concerned for gum arabic.
UV absorbance spectra for GA showed a hypochromic effect for all storage conditions. In turn, only GT stored under conditions 75% RH/25°C/3 days showed a hyperchromic effect.’’
5) ,,The references should be checked in terms of the style and cohesion.’’
Answer for Reviewer
As indicated by the Reviewer, the references was check in terms of the style and cohesion.

Reviewer 5 Report
Grammartical error and spelling mistakes were obtained. The authors should be revised.
line 74 Fig. 1 change to Figure 1
Author Response
REVIEWER 5
Comments and Suggestions, and Answers
We are very grateful for the valuable opinion and comments. All the suggested valuable changes have done in the corrected version of our paper.
1) ,,Grammartical error and spelling mistakes were obtained. The authors should be revised.’’
Answer for Reviewer
The new version of our manuscript, after the revision according to the Reviewers’ comments, was corrected by an English translator. The cases of spelling were corrected.
2) ,,line 74 Fig. 1 change to Figure 1’’
Answer for Reviewer
Has been corrected.

Reviewer 6 Report
In this study, authors studied the influence of storage conditions on the stability of gum arabic and tragacanth. It is well established that storage conditions can affect the action and stability of active pharmaceutical ingredients.
Authors studied the impact of UV-radiation, higher temperature and relative humidity on two polymers gum arabic and tragacanth using thermogravimetry (TG, c-DTA), colorimetric analysis, UV-Vis spectrophotometry, and optical microscopy as research methods. In their conclusion authors do not recommend storing polymers in the presence of UV radiation and increased temperature, especially in conditions of increased humidity in order to maintain their properties.
Overall, it’s a well-designed study with appropriate research methods. However, a major concern about reproducibility of results raised.
Indeed, authors did not indicate how many times analysis were performed (n?).
Are the results presented representative of several experiments? If not, I strongly recommend that authors ensure the reproducibility of experiments.
Minor concerns:
In the attached file, some minor spell checks have been pointed out.
Also, authors should harmonize English spelling (British vs. American) in the whole manuscript (e.g., analysis and analyzes).

Author Response
REVIEWER 6
Comments and Suggestions, and Answers
We are very grateful for the valuable opinion and comments. All the suggested valuable changes have done in the corrected version of our paper.
1) Indeed, authors did not indicate how many times analysis were performed (n?).
Are the results presented representative of several experiments? If not, I strongly recommend that authors ensure the reproducibility of experiments.
Answer for Reviewer
The measurements was performed three times for each sample. Information on statistical analysis has been added in section ,,Materials and Methods’’, point 3.9.
,,3.9. Statistical analysis
The measurements was performed three times for each sample. The results are presented as mean (±) standard deviations (SD). One-way ANOVA test were used to assess statistical significance. The significance level was assumed p<0.05. Statisitcal analysis was done using the program Statistica produced by TIBCO Software Inc. (Palo Alto, CA, USA).’’
2) Minor concerns:
In the attached file, some minor spell checks have been pointed out.
Also, authors should harmonize English spelling (British vs. American) in the whole manuscript (e.g., analysis and analyzes).
Answer for Reviewer
All minor remarks marked in the text by the Reviewer have been corrected. The new version of our manuscript, after the revision according to the Reviewers’ comments, was corrected by an English translator. The cases of spelling were corrected.

Round 2
Reviewer 2 Report
The authors are not responded well to previous concerns. This reviewer is not convinced regarding the use of spectrometry and colorimetric approaches for stability evaluation. A stability-indicating analytical method is one, which can detect the analyte in the presence of its degradation products. Spectrometry and colorimetry approaches do not have this property and hence not suitable for stability evaluation. In addition, chemical stability data is missing in the manuscript.
Reviewer 5 Report
Accept
Reviewer 6 Report
I went through the revised manuscript and it has been greatly improved. Authors have addressed all my concerns and the manuscript is now suitable for publication in its present form.